# Personal relative deprivation and pro-environmental intentions

**William J. Skylark**¹*, **Mitchell J. Callan**²

**1** Department of Psychology, University of Cambridge, Cambridge, United Kingdom, **2** Department of Psychology, University of Bath, Bath, United Kingdom

* w.j.skylark@cantab.net

**Data Availability Statement:** All relevant data are within the manuscript and its Supporting Information files.

**Funding:** The author(s) received no specific funding for this work.

## Abstract

Personal relative deprivation (PRD; the belief that one is worse off than other people who are similar to oneself) is associated with a reduced willingness to delay gratification, lower prosociality, and increased materialism. These results suggest that PRD may play a role in shaping people's willingness to act to protect the natural environment. We report 3 studies that investigate a possible link between PRD and pro-environmental intentions (ENV). Study 1 was an exploratory study using a US sample; Studies 2 and 3 were pre-registered replications using UK and US samples, respectively. In each study, participants self-reported PRD and ENV; they also indicated their subjective social status (where they come on a national "ladder" of social class) and reported their income, education, age, and gender/sex. All three studies found a negative correlation between PRD and ENV. However, multiple regression analyses in which ENV was regressed on PRD and all other variables simultaneously indicated that the unique effect of PRD was small and, for Studies 2 and 3, the 95% confidence intervals included zero. No other variable emerged as a clear unique predictor across all three studies. The data suggest that PRD may be associated with reduced intention to act pro-environmentally, but the causal status of this association, and its relationship to other demographic and social-status variables, remains a topic for further research.

## Introduction

Given the scientific consensus that humans need to adopt more sustainable patterns of behaviour and resource use [1,2], it is important to understand the drivers of sustainable behaviour (e.g., [3–6]). Correspondingly, much research has focussed on the links between pro-environmental behaviour and socio-economic variables such as income (e.g., [7–9]); other work has examined the contribution of psychological factors such as value orientation (e.g., [10]), social norms (e.g., [11]) and stress (e.g., [12]). The present paper investigates the links between pro-environmental intentions and a psychological variable that is related to, but distinct from, socio-economic status: personal relative deprivation.

Personal relative deprivation (PRD) is the state of believing oneself to be materially worse off than one's peers–that is, than people who are "like you"–with attendant feelings of

**Competing interests:** The authors have declared that no competing interests exist.

resentment and the belief that the world is not fair (see [13] for a review). PRD is distinct from objective indicators of socio-economic status (SES) such as annual household income and education: a person can be objectively wealthy but still feel deprived relative to others who are "like them". PRD is also distinct from subjective social status (SSS), as indexed by where a person places him- or herself on a national "ladder" of social class (e.g., [14]). Thus, a person can be objectively or subjectively "high status" on a national scale, but still feel resentment about how their material circumstances compare with those of their self-selected peers. Correspondingly, in previous work PRD has been found to show only modest correlations with objective and subjective socio-economic status (e.g., correlations ranging from -.18 to -.53 in [15]), and PRD predicts a range of social, economic and health outcomes after controlling for objective and/or subjective status (e.g., [15–18]).

Our interest in the possibility that PRD might be associated with environmental attitudes and behaviours was motivated by a consideration of some of the key elements of pro-environmentalism, and by recent work connecting these elements to PRD. First, acting to preserve the environment often involves focusing on long-term outcomes rather than immediate rewards (e.g., [19]). For example, the effects of rising $CO_2$ levels manifest over decades or centuries, and averting these harmful outcomes requires some immediate costs/losses (e.g., foregoing the ease of driving to work). Second, environmentalism often involves de-emphasising the acquisition of material possessions, with value instead being placed on activities and experiences that are less dependent on physical resources [20] (e.g., the production of impressive consumer products often involves heavy use of finite resources and harmful by-products, whereas a focus on spirituality or community makes fewer demands of the natural world). Third, although pro-environmentalism can be motivated by concerns for one's own well-being, it also often involves an element of pro-sociality [21]: environmental change affects everyone, and one's own actions can have harmful consequences for people thousands of miles away.

Previous work has found that PRD is associated with all three of these components of pro-environmentalism. First, higher PRD has been found to be associated with an increased focus on immediate financial rewards and steeper discounting of delayed outcomes [22–24] (see also [25,26]). Likewise, research suggests that PRD is associated with greater materialism–that is, people who feel deprived relative to similar others put more emphasis on the acquisition of money or material possessions [17,27,28]. Finally, there is evidence that PRD is negatively associated with prosociality [29]. For example, Callan et al. [15] found that PRD was negatively correlated with social values orientation [30] and with offers made in a dictator game (see also [31]). In many of these studies, PRD has been shown to have a causal effect on the outcome variable (i.e., manipulating people's sense of their relative deprivation shifts their willingness to delay gratification, to act for the benefit of others, or their enthusiasm for material possessions), and/or PRD has been found to predict the outcome variables over and above the effects of other indicators of social status (i.e., income, education, and subjective social status).

Based on this prior work, we hypothesized that there is an association between PRD and pro-environmentalism. To the extent that people higher in PRD have been found to be more focussed on immediate rewards, self-interest, and material values, they might be expected to show reduced pro-environmental intentions and, ultimately, fewer pro-environmental behaviours.

We conducted 3 studies to test this possibility, focussing on pro-environmental intentions. (We acknowledge from the outset that such intentions will imperfectly translate into behaviours [32]; nonetheless, intentions are often an important antecedent of behaviour [33], are somewhat easier to measure, and might be expected to be more closely related to the psychological processes shaped by PRD.) Study 1 was an exploratory study (i.e., the analysis plan was not rigidly specified in advance of data collection [34]) using a US sample; Studies 2 and 3

were pre-registered replications that used UK and US samples, respectively. In each case, participants self-reported their level of PRD and their intentions to engage in pro-environmental behaviours (ENV); they also indicated their age, gender (or sex), level of education and annual household income, and their subjective social status (SSS). We examine whether PRD is correlated with ENV, and whether PRD uniquely predicts ENV after controlling for the other variables. In all cases, we are interested in estimating effect sizes and the uncertainty attached to those estimates (i.e., the width of confidence intervals) rather than on binary "effect vs no effect" decisions.

## Methods

Studies 2 and 3 were pre-registered at https://aspredicted.org/as2fq.pdf and https://aspredicted.org/gt4n9.pdf, respectively.

### Ethical approval

Study 1 was approved by the Department of Psychology Research Ethics Committee at the University of Bath; Studies 2 and 3 were approved by the Department of Psychology Ethics Committee at the University of Cambridge (REF: 2021/55). Participants in Study 1 provided consent by clicking a "continue" button at the end of an Information sheet; those in Studies 2 and 3 provided consent by selecting yes/no answers to a series of consent questions after an Information Sheet; only those who answered "yes" to all questions were able to proceed to the study.

### Measures, design and procedure

The studies were conducted online using the Qualtrics survey software (www.Qualtrics.com). After an initial information/consent sheet, participants completed the following three measures, in random order with one measure per page:

**Subjective Socioeconomic Status (SSS).** Participants were presented with an image of a 10-rung ladder and asked to think of it as representing the social hierarchy of the United States (Studies 1 and 3) or United Kingdom (Study 2), with the people who have the most money, highest education, and best jobs at the top. Participants were asked to indicate where they fit into this social ladder, with responses coded 1–10.

**Personal Relative Deprivation (PRD).** Participants completed the 5-item PRD scale reported by Callan et al. (2011); example item: "I feel deprived when I think about what I have compared to what other people like me have"; participants indicated their agreement on a 6-point scale (Strongly disagree to Strongly agree, coded 1–6, with reverse-coding of 2 items). The average response was used as the index of PRD.

**Pro-environmental Intentions (ENV).** We used the 12-item questionnaire reported by Bain et al. (2016) to probe the participant's intentions to engage in a set of sustainable/pro-environmental behaviours over the next 12 months. Example items include: "Buy products with less packaging" and "Eat food which is locally grown or in-season". Participants responded on a 5-point scale (Not at all likely to Very Likely, coded 1–5) with an additional option "Not Applicable" to be selected if it was not possible for the participant to engage with an activity. Between items 5 and 6 there was an attention check ("To check that people are reading, please select "Somewhat unlikely" for this statement"). The mean of the participant's responses (excluding cases where they selected "Not Applicable" and the attention check item) was used as the index of pro-environmental intentions.

After these three measures, participants completed a page of demographic questions that asked: (1) their annual household income (before taxes), with response options "less than

$10,000", "10,001 to $20,000" and so on in steps of $10,000 up to a top category ("More than $150,000") (for Study 2, the income options were expressed in pounds Sterling rather than dollars). Responses were coded using the midpoint of each category divided by 1000 (i.e., 5, 15, 25. . .) with Parker and Fenwick's [35] median-based estimator used for the top category (e.g., [36]); (2) their highest level of educational attainment (response options: did not finish high school; high school graduation; college graduation; postgraduate degree, coded 1–4; for Study 2, these options were replaced by 6 options suitable for a UK sample); gender (male, female, non-binary, coded -0.5, +0.5, 0; for Studies 2 and 3 this was replaced by "sex" with response options male, female, prefer not to say, coded -0.5, +0.5, and 0 respectively); age (indicated by a slider ranging from 0 to 100 in Study 1, and by a text box that accepted only numeric input in Studies 2 and 3); and, for Study 1 only, current US state of residence (indicated by selecting from a drop-down list of states; these responses were not for analysis). For Studies 2 and 3, a page was added after the demographic questions asking whether this was their first attempt at the survey.

In Studies 2 and 3, all questions required a response before the participant could progress. In Study 1, the SSS, PRD, ENV and State-of-Residence questions did not require responses, and a glitch meant that participants could select more than 1 rung of the SSS ladder (although hardly anyone did). In all studies, the final participant samples comprise people who answered all questions and only indicated one rung on the SSS ladder. After the final question, participants were shown a debriefing sheet.

The precise wording of the items and response options are provided in the Supplementary Materials (S1 Appendix).

## Participants

For Study 1, participants from the USA were recruited via Amazon's Mechanical Turk (www. mturk.com). For Study 2, the participants were UK-residents whose first language was English and who were not currently students, recruited via Prolific (www.prolific.co); Study 3 was like Study 2 except the participants were resident in the USA. We only analysed data from participants who answered all questions (and who selected just 1 rung of the SSS ladder), passed the attention check, and who were not flagged as having potentially started the study (or, for Studies 2 and 3, one of the earlier studies) before. Full details of the sampling/data screening procedure are provided in the Supplementary Materials (S1 Appendix) and, for Studies 2 and 3, followed the pre-registered plans. The final samples comprised 308, 409 and 423 participants for Studies 1, 2, and 3, respectively, and are described in Table 1. These samples sizes give, respectively, 70.2%, 81.9%, and 83.1% power to detect a small ($r = .141$, accounting for 2% of the variance) correlation between PRD and ENV, and 97.8%, 99.6% and 99.7% power to detect an association $r = .224$ (accounting for 5% of the variance) [37].

## Data analysis

Data analysis was conducted in R version 4.0.3 [39] using the tidyverse package version 1.3.0 [40], the parameters package version 0.11.0 [41], and other packages described in the Results section. The data for all 3 studies are available in the Supplementary Materials (files S1–S3 Data; an explanation of the contents of the data files is provided in S1 Appendix), along with a copy of the analysis script (S1 Script). We examined the correlations between variables and regressed ENV on PRD, SSS, age, gender/sex, income, and education. Unless otherwise noted, all confidence intervals are 95% intervals. As described in the pre-registrations, we base our inferences on the CIs, but for completeness we also report the $p$-values for the associated two-tailed tests of the key effects. Further details are provided in the Results section.

**Table 1. Sample characteristics.**

|  | Study 1 (N = 308; 194 male, 111 female, 3 Other) | | | |
|---|---|---|---|---|
|  | Min | Max | Mean (SD) | Cronbach's α |
| AGE | 20 | 65 | 35.89 (10.56) | — |
| EDU | 2 | 4 | 2.71 (0.62) | — |
| INC | 5 | 185.85 | 55.68 (40.18) | — |
| SSS | 1 | 9 | 4.72 (1.81) | — |
| PRD | 1.00 | 6.00 | 3.21 (1.19) | .85 |
| ENV | 1.00 | 5.00 | 3.78 (0.74) | .83 |
|  | Study 2 (N = 409; 154 male, 253 female, 2 prefer not to say) | | | |
|  | Min | Max | Mean (SD) | Cronbach's α |
| AGE | 18 | 79 | 39.01 (12.48) | — |
| EDU | 1 | 6 | 3.69 (1.00) | — |
| INC | 5 | 160.71 | 45.97 (29.43) | — |
| SSS | 1 | 9 | 5.34 (1.55) | — |
| PRD | 1.00 | 5.80 | 2.94 (0.94) | .83 |
| ENV | 1.92 | 5.00 | 4.06 (0.51) | .75 |
|  | Study 3 (N = 423; 213 male, 205 female, 5 prefer not to say) | | | |
|  | Min | Max | Mean (SD) | Cronbach's α |
| AGE | 18 | 74 | 36.54 (11.45) | — |
| EDU | 1 | 4 | 2.92 (0.72) | — |
| INC | 5 | 189.71 | 70.92 (47.75) | — |
| SSS | 1 | 9 | 5.24 (1.81) | — |
| PRD | 1.00 | 6.00 | 3.04 (1.10) | .86 |
| ENV | 1.08 | 5.00 | 3.81 (0.67) | .81 |

Note: For Study 2, we pre-registered that we would compute $\omega_h$ as well as Cronbach's α (computed using the psych package version 2.0.12 [38]); however, the software reported warnings during the computation for PRD, so we report the values without further comment: For PRD, $\omega_h$ = .62, for ENV, $\omega_h$ = .57.

## Results

### Correlation analyses

Fig 1 plots the Pearson correlation coefficients and 95% confidence intervals for the correlation matrix between ENV, PRD, SSS, income (INC), education (EDU), age, and gender/sex (SEX). (The numeric values plotted in this figure are provided in the Supplementary Materials, S1 Table). For all three studies, higher personal relative deprivation is associated with reduced pro-environmental behavioural intentions: Study 1: $r$ = -0.266, CI = [-0.367, -0.159], $p < .001$; Study 2: $r$ = -0.141, CI = [-0.235, -0.045], $p$ = .004; Study 3: $r$ = -0.125, CI = [-0.218, -0.030], $p$ = .010. Notably, the association was stronger in Study 1 than in the two subsequent pre-registered studies. Fig 1 also indicates that, as would be expected, personal relative deprivation, subjective social status, and objective status indicators are mutually inter-related, and that SSS is positively associated with pro-environmental intentions across all three studies. Replicating previous work [42], all three studies also find that older participants report lower PRD than do younger ones, although the CIs for this effect include zero for Study 1.

We also computed the Kendall's correlation matrix (for Studies 2 and 3, this robustness check was pre-registered; the CIs were computed using the DescTools package version 0.99.40 [43]). The pattern of results was very similar to the Pearson's correlation matrix. In particular, pro-environmental intentions were negatively associated with PRD in all three studies: Study

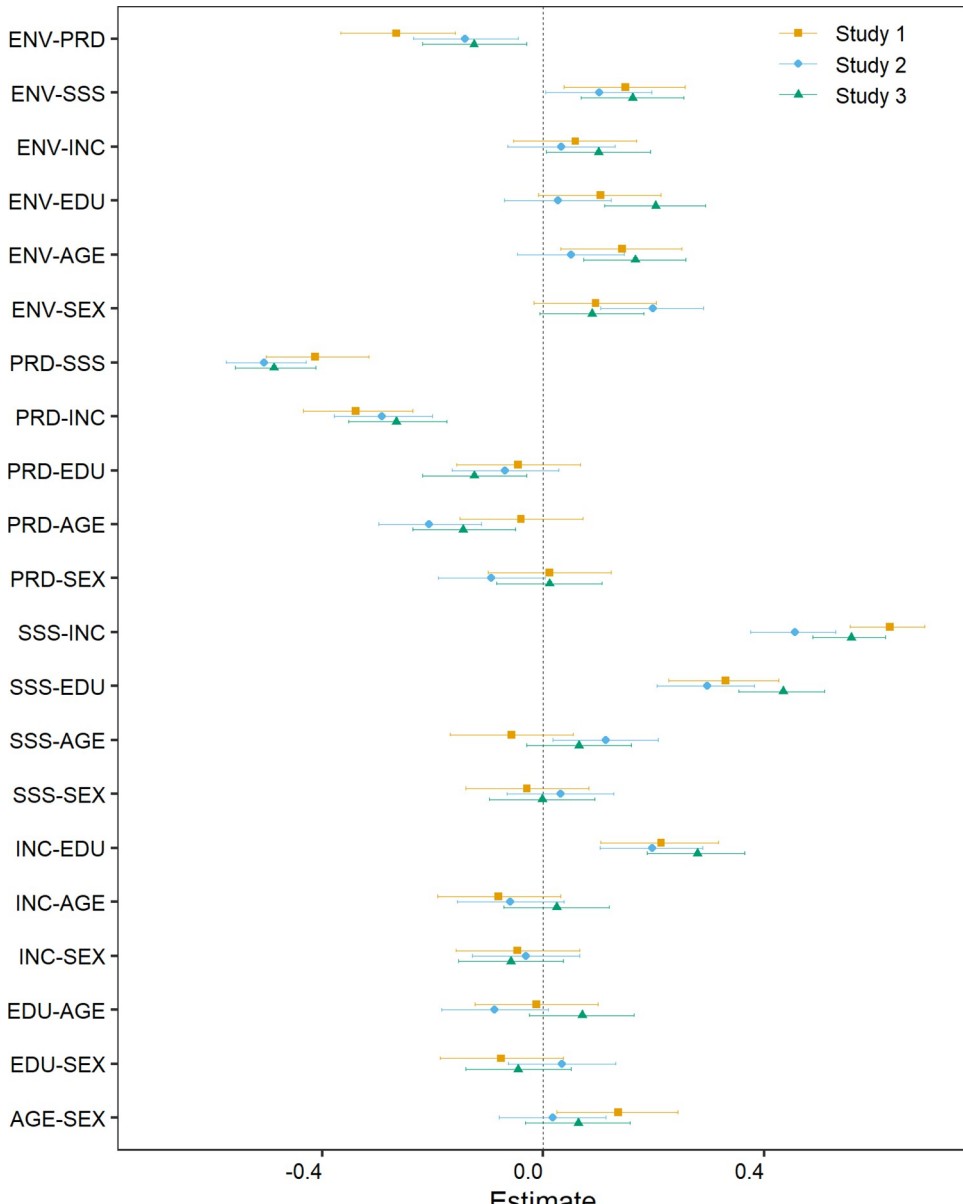

**Fig 1. Pearson correlations.** Each point shows the correlation between the pair of variables listed on the y-axis; the error bars show 95% confidence intervals.

1: tau = -0.175, CI = [-0.254, -0096], $p < .001$; Study 2: tau = -0.099, CI = [-0.165, -0.032], $p = .004$; Study 3: tau = -0.073, CI = [-0.138, -0.008], $p = .031$. (The full set of Kendall's correlation results are provided in the Supplementary Materials, S1 Table).

## Regression analyses

To establish whether PRD uniquely predicts ENV after controlling for other variables, we regressed ENV on PRD, SSS, income, education, age, and sex (or gender). All variables apart from sex/gender were z-scored (so the regression coefficient indicates the change, in SD units, associated with a one-standard-deviation increase in the predictor). The regression coefficients

for all three studies are plotted in Fig 2 (the numeric values used in this plot are provided in the Supplementary Materials, S2 Table); the goodness of fit statistics are: Study 1, $R^2 = .109$, $R^2_{adj} = .091$; Study 2, $R^2 = .057$, $R^2_{adj} = .043$; Study 3, $R^2 = .081$, $R^2_{adj} = .068$. In all three studies, the coefficient for PRD is negative, but for Studies 2 and 3 the effect is very small and the 95% CIs include zero. (Numerically, the coefficients are: Study 1, B = -0.255, CI = [-0.374, -0.136], $p < .001$; Study 2, B = -0.098, CI = [-0.211, 0.016], $p = .091$; Study 3, B = -0.060, CI = [-0.168, 0.047], $p = .270$.)

We ran additional regression analyses to probe whether the results of our initial analysis were the result of particular analytic decisions [44] (these additional analyses were pre-registered for Studies 2 and 3). Specifically, we re-ran the regressions using the robust regression

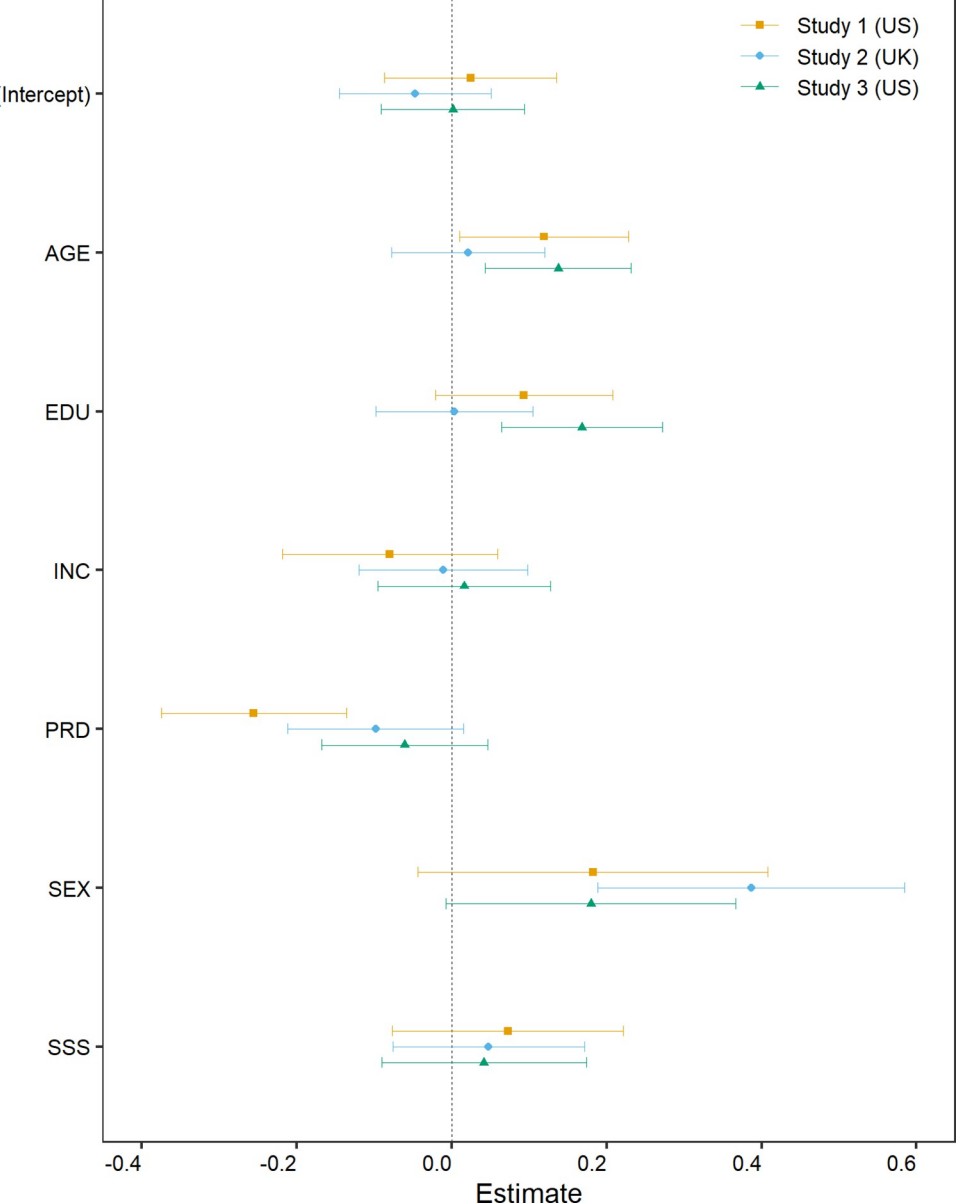

**Fig 2. Regression results.** Each point shows the regression coefficient for the predictor listed on the y-axis. The error bars show 95% confidence intervals.

function lmrob from the robustbase package version 0.93–6 (with setting ="KS2014" [45]), and then repeated both the OLS and robust regression analyses using (a) log-transformed income (in place of raw income; the logarithm was to base 10, and again the values were z-scored), (b) a composite measure of socio-economic status, formed by z-scoring education and income, averaging them, and then z-scoring (in place of separate indicators for income and education), and (c) coding education as a factor with weighted effect coding using the wec package 0.4–1 [46,47]. These different approaches to objective socio-economic indicators reflect the diverse approaches taken by researchers in previous investigations of PRD and SSS (e.g., [7,48–50]). For all three studies, the pattern of results for these alternative regression analyses were very similar to those shown in Fig 2 (the full results are reported in the Supplementary Materials, S2 Table). In particular, for Study 1, the CIs for the effect of PRD excluded 0 for all analyses, and for Studies 2 and 3 the CIs included zero for all analyses except the robust regression analysis with education treated as a factor for Study 2, where the CI just excluded zero (B = -0.111, CI = [-0.221, -0.002], $p$ = .047).

## Meta-analyses

The pre-registration for Study 3 specified that we would meta-analyse the Pearson correlations between PRD and ENV and the semi-partial correlations between PRD and ENV from the regression analyses [51,52]; we did this using the metafor package version 2.4–0 [53]. Internal meta-analyses can be prone to bias [54]; for example, in our case Study 1 was not pre-registered and, although there was no "optional stopping" or selective reporting of measures, it might nonetheless have (inadvertently) involved analytic decisions that inflated the effect. One approach would be to exclude Study 1 altogether; we thought it better to incorporate all of the data and to focus on 99% confidence intervals to offer some protection against the possible bias introduced by including Study 1, whilst also acknowledging that the resulting effect-size estimate might still be an overestimate. For completeness, we also report the associated $p$-values; given our focus on 99% CIs, the appropriate criterion for "significance" for the population effect estimates would be .01.

For the correlation meta-analysis we used Fisher's r-to-z transformation and then back-transformed to the original scale when reporting the results. A random effects meta-analysis estimated the population effect as $r$ = -.174, 99% CI = [-.282, -.062], $p$ < .001. (As pre-registered, we also report the test for heterogeneity: Q(df = 2) = 4.341, $p$ = .114.) The pre-registration said that we would also conduct a fixed effects meta-analysis: the estimated population effect is $r$ = -.169, 99% CI = [-.243, -.094], $p$ < .001. Thus, the meta-analysis indicates a negative association between PRD and ENV, although the central estimate of the effect is small and the confidence intervals come close to zero.

For the random effect meta-analysis of semi-partial correlations, the test for heterogeneity yielded Q(df = 2) = 7.017, $p$ = .0299. The estimate of the population semi-partial correlation was $sr$ = -.119, 99% CI = [-.257, .020], $p$ = .027 (not "significant" given our pre-registered focus on 99% CIs). As pre-registered, we applied the same meta-analytic approach to the estimated effect of PRD from the other regression analyses (i.e., the OLS and robust regressions using different approaches to income and education): the results were virtually identical (see Supplementary Materials, S3 Table).

## Discussion

These studies found some support for the idea that higher personal relative deprivation is associated with lower pro-environmental behavioural intentions: in Study 1, PRD was negatively correlated with ENV, and this pattern was reproduced in two pre-registered replications

(Studies 2 and 3). However, the effect was small. Notwithstanding the limitations of internal meta-analyses [54], the central estimate of the correlation obtained from random-effects meta-analysis indicates that PRD accounts for approximately 3% of the variance in ENV. The modest effect is perhaps unsurprising given the complexity and multiply-determined nature of environmental attitudes and behavioural intentions (e.g., [55–57]). To help contextualize the current results, we note that Study 1 of Kim et al. [17] reports a correlation between PRD and scores on the Material Values Scale of $r = .49$, whereas Study 1 of Callan et al. [15] reports a correlation between PRD and scores on the Social Values Orientation scale [30] of $r = -.117$ (i.e., smaller than the meta-analytic estimate obtained in the current work).

After controlling for the other demographic and status variables, the association between PRD and ENV was not reliably different from zero: the effect found in Study 1 did not replicate in the two pre-registered studies (that is, the effect was directionally replicated but the CIs included zero), and the meta-analytic central estimate of the unique effect of PRD was very small with 99% confidence intervals that stretched from a small negative effect to a tiny positive effect. This situation contrasts with that reported by Callan et al.'s aforementioned study of the link between PRD and prosociality [15]: in that paper, PRD and subjective indicators of social class acted as mutual suppressors, such that the association between PRD and prosociality was strengthened after controlling for SSS and income/education. However, in that work the zero-order correlation between SSS and various measures of prosociality was typically very small and negative, whereas in the present studies SSS showed a consistent positive association with pro-environmental intentions.

The difference between the zero-order correlation and multiple regression results in the current work suggests that the correlation between PRD and pro-environmental intentions is due to a confound. Inspection of Fig 1 suggests SSS as an obvious candidate: it is moderately correlated with PRD ($r$ typically about -0.4) and is the only variable apart from PRD that was consistently correlated with ENV. However, as shown in Fig 2 the multiple regression results suggest that SSS did not uniquely predict ENV in any of the three studies (if anything, PRD appears to be the more important variable). In fact, none of the predictors emerged as a consistent "winner" in the regression analyses, and there was considerable variability in the results between studies. For example, age was positively associated with pro-environmental intentions in the US samples (Studies 1 and 3) but the effect was close to zero for the UK (Study 2); the effect of education shows a similar pattern. Conversely, females were more pro-environmental than males in the UK sample, but the effect seems to be smaller (and the CIs include zero) in the US samples. Notably, there are also indications that the associations between PRD and these other variables differ across samples. For example, sex/gender shows virtually no association with PRD in the US samples (Studies 1 and 3) but for the UK sample there is some indication that males report higher PRD than females; and for Studies 2 and 3, PRD is higher among younger participants but this effect is notably weaker (and not "significant") in Study 1. We cannot tell whether these differences reflect sampling error or structural differences between the sampled populations (e.g., MTurk vs Prolific, US vs UK). Nonetheless, they suggest that, for the populations sampled here, none of the demographic or status variables act as a consistent "main driver" of pro-environmental intentions.

It is important to note several limitations of our work. First, although our samples are quite diverse, they were not nationally representative, and they were drawn from just two countries. A multinational study with representative samples would help clarify the generality (and potential moderators) of the effects reported here. A second issue is that we used a single measure of environmental intentions; other scales or observations of actual behaviour might yield different results (e.g., [58,59]). Notably, the scale we used reports intentions for future behaviours to reduce environmental impact, but a person who is highly pro-environmental might

already be maximally performing that behaviour (e.g., they may not be able to increase the amount they recycle if they recycle everything already). More generally, intentions are often only a weak predictor of behaviour (see e.g., [32] for a review). Beyond this, there are additional control variables, moderators, and mediators that could be considered–for example, we have not examined the possibility that the effects of relative deprivation or social status might vary by gender or national culture [60], and we took no account of political orientation. Finally, our study is cross-sectional and correlational, so it would be fruitful to perform longitudinal research to see whether relative deprivation at time 1 predicts behaviour at time 2, and, perhaps, to employ experimental approaches in which participants' sense of relative deprivation is manipulated by false-feedback or induced social comparison [22,27,48].

Given that these studies are (as far as we know) the first to investigate the potential link between personal relative deprivation and pro-environmental intentions, we think it best to avoid excessive speculation about possible interpretations of our findings at this point, but Reviewers suggested that it might be helpful to comment on the possible practical significance and broader theoretical implications of our findings, so we conclude with two brief remarks.

First, our results tentatively suggest that, while interventions intended to reduce people's experiences of PRD (for example, by encouraging lateral or downward social comparisons [31,48]) can have positive consequences, such interventions might be of limited value in encouraging pro-environmentalism.

Second, our results might indicate a more general principle regarding when PRD will/will not be a substantive predictor of attitudes and behaviours. The present studies were motivated by recent evidence that prosociality, desire for immediate rewards, and material values are all associated with PRD. Although these variables are relevant to environmentalism, they are also directly linked to the core elements of personal relative deprivation–namely, the allocation of material resources between people. In contrast, environmentalism substantially concerns the relations between humans and the natural (non-human) world. Our results therefore accord with the possibility that PRD is primarily predictive of attitudes and behaviours that directly concern social relations and resource distribution, rather than those which are non-social (but see [16]). Additionally, we note that PRD is defined by feelings of resentment at being treated unfairly [13,22,48]. This raises the possibility that a link between PRD and environmentalism might emerge more clearly if environmental issues are couched in terms of justice and fairness (for example, between different regions of the world or age groups), and therefore more directly linked to the core cognitions and emotions involved in PRD. These possibilities, coupled with the methodological refinements noted above, could serve as useful directions for future work.

## Supporting information

**S1 Table. Full set of correlation results.** In this file, Parameter indicates the pair of variables whose correlation is being tested; Coefficient indicates the Pearson's *r* or Kendall's tau value; CI_low and CI_high are the 95% confidence intervals; p is the p-value; Analysis indicates whether the correlation is Pearson or Kendall; Study indicates the Study.
(CSV)

**S2 Table. Full set of regression results.** In this file, Analysis indicates the regression model and fitting procedure. (The models are described in the main text): Analyses 1–4 use OLS regression; Analysis 1 is the model reported in the main text; Analysis 2 uses z-scored log-transformed income; Analysis 3 uses the SES composite; Analysis 4 is like model 1 but with education coded as a factor. Analyses 5–8 are the same was Analyses 1–5, respectively, but are fit with robust regression. Parameter indicates the predictor variable; Coefficient indicates the

regression coefficient; SE indicates the standard error of the estimate; CI_low and CI_high are the lower and upper 95% confidence intervals; t, df_error, and p are the associated t-value, degrees of freedom, and p-value; rsq and adjrsq are the R-sq and adjusted R-sq for the overall model (so they are the same for all parameters for a given Analysis); Study indicates the Study.
(CSV)

**S3 Table. Full set of results of meta-analyses of semi-partial correlations.** In this file, Analysis indicates the regression analysis to which the meta-analysis relates, with the same naming as for S2 Table. Q is the test statistic for the test of heterogeneity (with 2df); Qp is the associated p-value; beta is the estimated effect; SE is the associated standard error; CI_low and CI_high are the 99% confidence intervals for the effect; p is the associated p-value.
(CSV)

**S1 Appendix. Details of study materials, participant sampling, and explanation of the data files.**
(PDF)

**S1 Data. Data for Study 1.** The columns are explained in S1 Appendix.
(CSV)

**S2 Data. Data for Study 2.** The columns are explained in S1 Appendix.
(CSV)

**S3 Data. Data for Study 3.** The columns are explained in S1 Appendix.
(CSV)

**S1 Script. Analysis script for Studies 1–3.**
(R)

## Author Contributions

**Conceptualization:** William J. Skylark, Mitchell J. Callan.

**Data curation:** William J. Skylark, Mitchell J. Callan.

**Formal analysis:** William J. Skylark, Mitchell J. Callan.

**Investigation:** William J. Skylark, Mitchell J. Callan.

**Methodology:** William J. Skylark, Mitchell J. Callan.

**Writing – original draft:** William J. Skylark.

**Writing – review & editing:** Mitchell J. Callan.

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
