## [Decision Letter · Decision Letter 0]

27 Aug 2021

PONE-D-21-19291

Personal relative deprivation and pro-environmental intentions

PLOS ONE

Dear Dr. Skylark,

Thank you for submitting your manuscript to PLOS ONE. After careful consideration, we feel that it has merit but does not fully meet PLOS ONE’s publication criteria as it currently stands. Therefore, we invite you to submit a revised version of the manuscript that addresses the points raised during the review process.

As you can see, the reviewers find the theme of your manuscript interesting, but point to shortcomings and weaknesses that need to be addressed and remedied. I am also in line with their insightful comments and suggestions. When resubmitting please indicate in detail how the revised version addresses all the referees’ concerns and concomitant suggestions.

We look forward to receiving your revised manuscript.

Kind regards,

Athina Economou

Academic Editor

PLOS ONE

Journal Requirements:

Reviewers' comments:

Reviewer's Responses to Questions

**Comments to the Author**

1. Is the manuscript technically sound, and do the data support the conclusions?

Reviewer #1: Yes

Reviewer #2: Partly

Reviewer #3: Yes

2. Has the statistical analysis been performed appropriately and rigorously? 

Reviewer #1: Yes

Reviewer #2: Yes

Reviewer #3: Yes

3. Have the authors made all data underlying the findings in their manuscript fully available?

Reviewer #1: Yes

Reviewer #2: Yes

Reviewer #3: Yes

4. Is the manuscript presented in an intelligible fashion and written in standard English?

Reviewer #1: Yes

Reviewer #2: Yes

Reviewer #3: Yes

5. Review Comments to the Author

Reviewer #1: Thank you for providing me with the opportunity to review this paper for PLOSOne.

The authors explore an interesting question in their studies, that being, the extent to which people’s motivations to engage in pro-environmental behaviour may be related to their sense of personal relative deprivation in comparison to their similar peers.

The authors provide a reasonable (albeit rather slim) theoretical and empirical rationale for their research question, and their empirical work appears to have been conducted in a careful and rigorous fashion (and is reported in a refreshingly transparent way).

Whilst it is difficult to find fault with the scientific validity of the empirical work, I did feel that the authors could have done a little more to engage with the potential implications of their findings. Obviously, the results were not as clean cut as one might have hoped, or at least the effect sizes were rather small. However, given that the authors do make a general conclusion that PRD seems to be to some extent negatively related to pro-environmental behaviour, it would be good for them to say a bit more about a) why they think this might be, and b) what the potential theoretical implications might be of this being the case, and c) what some of the practical implications of this might be.

Re a), the authors suggest in the introduction that possible mediators of the proposed relationship might be greater focus on immediate financial rewards, greater materialism and lower prosociality. Given this, I was slightly surprised to not see any attempt to measure these proposed mediators across any of the reported studies.

Re b), is there anything that the findings reported here might tell us about broader theoretical questions, such as the nature of pro-environmental behaviour or indeed the nature of PRD?

Re c), if higher levels of PRD might indeed dampen down pro-environmental intentions, then what might this mean for those individuals, organisations or government bodies who seek to increase engagement with pro-environmental behaviour?

At present, the ending of the paper simply states limitations of the study. Stating limitations is important and useful. However, I think the paper would benefit greatly from the addition of a final paragraph that brings things back to considering the wider research problem that was laid out at the start of the introduction, and that considers issues a, b and c outlined above.

As a very minor point, there appeared to be a typo on page 16 (“The modest effect is perhaps unsurprising give the complexity and multiply-determined nature of environmental attitudes and behavioural intentions”)

Additionally, in the discussion the authors state that “we have not examined the possibility that the effects of relative deprivation or social status might vary by gender”. Surely the authors have the data at their disposal to test this though?

Reviewer #2: The manuscript “Personal relative deprivation and pro-environmental intentions” examines the correlation between personal relative deprivation and pro-environmental intentions. In three studies, the authors find a weak but significant effect.

The topic is very interesting, and I believe that much research is needed in this domain. Moreover, I think the sample sizes are very nice. However, while I think the aim of the manuscript is important, I have some concerns about the contribution of the manuscript, as I outlined below:

1. The paper lacks a theoretical framework. The authors should broaden the theoretical framework and better explain why PRD should affect pro-environmental intentions and ultimately behavior (although it is important to note that the current studies only measured intentions and not behavior).

2. Relatedly, the authors talk about pro-environmental behavior but only measure intentions. Therefore, the gap between intentions and behaviors should be discussed both in the Introduction and Discussion sections.

3. The results are very weak. While I think that even null results might be informative (and thus, even when hypotheses are not confirmed, the data could be informative and important), I am not sure what the current paper’s theoretical and practical contribution is.

4. Why Study 1 was exploratory? Exploratory in what sense?

Minor comments:

1. Table 1 seems redundant. The Min, Max, and mean values of the variables are not very informative. I think it could be moved to the supplemented materials, while the Cronbach alphas could be indicated in the method section

2. The manuscript includes some typos (e.g., PRDS instead of PRD in several locations). Please correct

I hope the authors will find these comments useful as they continue with this interesting project

Reviewer #3: This manuscript examines the impact of personal relative deprivation (PRD) on willingness to act in ways to protect the natural environment. It compares the impact of PRD to other frequently investigated indicators of social (class) standing, and thus offers insight into how perceptions rather than more objective indicators affect behavioral intentions. It seems to meet the criteria for publication in Plos One (https://journals.plos.org/plosone/s/criteria-for-publication), e.g., the research is original, not published elsewhere, meets high methodological, statistical, and (largely) ethical standards, is written clearly. Yet I think that there are ways to improve the manuscript.

Despite the authors’ commendable endeavors to secure results from three samples (two of which were registered), the manuscript could have more forcefully communicated why studying PRD in relationship to environmental behavior intentions was important. It seemed that they just tacked an additional variable on to a long list of other status indicators known to be associated with such behaviors. While the authors tell us how PRD is related to materialism, prosociality, and so forth, they simply conclude that it “might be associated with reduced pro-environmental behavioral intentions.” How will knowing of this association affect expectations for such behavior? Could the argument be couched so that a direction of the association be indicated? Given the other more objective indicators of social standing, should PRD matter more or less? In other words, the justification for the study and the theoretical argument shaping the analyses struck me as thin and could certainly be bolstered.

Methodologically the research seems sound. I was, however, concerned that participation in Studies 2 and 3 was not fully voluntary as “all questions required a response before the participant could progress.” Typically, study volunteers are allowed to skip questions thereby upholding their right to continue or discontinue with the study. (I recognize that the studies under went ethical review, but this requirement would have been disallowed by my university’s Institutional Review Board.) I admit that I longed for a regression table rather than the figures of confidence intervals as I find interpreting coefficients easier. The authors note on page 9, line 231 that conducting a fixed effects meta-analysis was a mistake, but do not explain why it was a “mistake” except to say a random effects model makes more sense. Is such a passage necessary?

I had hoped that the discussion might do more than simply recount the findings – which demonstrate that PRD really had little effect, especially in the context of the other demographic and social standing variables. Plos One offers the possibility of publishing such null effects, but I nonetheless wondered about the meaning of this pattern of findings in terms of understanding what counts in producing pro-environmental behaviors. The authors do note that other variables could be considered. Given that much previous research has considered many other factors (social, cognitive, emotional), I still wonder about the contribution even though such concerns seem outside of the given review criteria.

6. PLOS authors have the option to publish the peer review history of their article (what does this mean?). If published, this will include your full peer review and any attached files.

Reviewer #1: No

Reviewer #2: No

Reviewer #3: No

---

## [Author Response · Author response to Decision Letter 0]

8 Sep 2021

Please see the attached file "Response to Reviewers".

---

## [Editor Report · Decision Letter 1]

26 Oct 2021

Personal relative deprivation and pro-environmental intentions

PONE-D-21-19291R1

Dear Dr. Skylark,

We’re pleased to inform you that your manuscript has been judged scientifically suitable for publication and will be formally accepted for publication once it meets all outstanding technical requirements.

Kind regards,

Athina Economou

Academic Editor

PLOS ONE

Additional Editor Comments (optional):

Many thanks for submitting your manuscript to the PLOS One Journal. Please accept my apologies for the delay in the process. Given the referees’ comments and your revised manuscript, I see no reason to proceed to a second round of reviews, since you addressed all reviewers' comments adequately. 
---

## [Editor Report · Acceptance letter]

10 Nov 2021

PONE-D-21-19291R1 

Personal relative deprivation and pro-environmental intentions 

Dear Dr. Skylark:

I'm pleased to inform you that your manuscript has been deemed suitable for publication in PLOS ONE. Congratulations! Your manuscript is now with our production department. 

Kind regards, 

on behalf of

Dr. Athina Economou 

Academic Editor

PLOS ONE